# Management of the Russian Interregional Investment Distribution Using the Autonomous Expenditure Multiplier Model

Sergey Nikolaevich Silvestrov *, Sergey Alekseevich Pobyvaev, Stanislav Borisovich Reshetnikov and Dmitrii Vladimirovich Firsov

Institute for Economic Policy and Problems of Economic Security, Financial University under the Government of the Russian Federation, 125167 Moscow, Russia; pobyvaev@inbox.ru (S.A.P.); stanislav-reschetnikov@bk.ru (S.B.R.); dmitrii-firsov@list.ru (D.V.F.)
* Correspondence: silvestrov-sergey@mail.ru

**Abstract:** An effective and competitive investment policy requires improvements to the existing tools. The ongoing COVID-19 crisis requires understanding as to how the recovery processes should be implemented. This study aims to develop a model for determining the autonomous expenditure multiplier (AEM) values, considering the investment accelerator action. The scientific novelty consists of proving that the AEM is not only an effects enhancer of the government and private investment, but also a tool to specify on the regional industrial map of Russia where investment projects will allow significant economic growth. The work's practical significance consists of determining the possibility of applying the AEM as a tool to improve investment efficiency. The key research method was paired linear regression analysis. Based on the developed model, the AEM values for the economies of the five Central Federal District regions are calculated. Additionally, authors provide an explanation on how AEM values correlate to regional economic specialization. For example, atypically low AEM values for Moscow can be explained by high daily workforce movement among Moscow and the Moscow region. The information support difficulties of the proposed model are defined, and the directions to overcome them are proposed. Empirical results show significant differences in the AEM values of the researched regions, and that the AEM as a management tool for interregional investment distribution will help to invest the limited resources of both the state and private businesses more effectively. Additionally, authors establish that the achieved results fall in line with real macroeconomic situations within the regions, which proves that the proposed model reflects real world processes. Primary beneficiaries and end users of the study are government agencies, state-owned corporations, and members of broader scientific community.

**Keywords:** investment; Russia; regions; multiplier; accelerator; autonomous expenditure



## 1. Introduction

The modern Russian economy is going through a rather difficult stage of its development. Its complexity lies in the simultaneous action of several powerful factors that negatively affect the dynamics of the economic development of the Russian Federation (The World Bank 2021). These include social, political, and economic factors. A powerful social factor is the COVID-19 pandemic, which affects the Russian economy in two ways: a decline in economic activity due to the necessary anti-COVID measures and the fall in oil prices due to the decline in global economic activity (Oxenstierna 2021). At the end of 2020, oil prices resumed their growth, but the trend of the world economy decarbonization (political factor), expressed in practice in the development of electric transport and solar and wind energy, along with the planned introduction of a hydrocarbon import tax in the European Union in 2022, poses a serious barrier to the further growth of oil prices

(Aylor et al. 2020). The equal barrier is shale production (technological factor), which becomes more active when oil prices rise above USD 40 per barrel and, consequently, creates conditions for a further decline in oil prices (Ali et al. 2022).

A significant social factor is the shrinking and aging of the Russian population. An important political factor is the sanction pressure on the Russian Federation, including the negative impact on the country's image in the eyes of potential investors.

The specifics of the Russian economy organization and a succession of external scientific, technical, and anthropogenic factors force us to take a new look at the processes of fixed assets and capital reproduction. Economic development processes require us to understand the relationship between the investment regional distribution and tools for assessing their effectiveness. Modern tools of investment economic evaluation are characterized by a common weakness in determining the directive relationship between the change in equilibrium gross domestic product (GDP) and the change in investment activity in the region (Helal 2019; Numbu and Belyaeva 2021; Gokmen 2021).

Russia is one of the few major countries in the world whose economic and societal realities were formed not by a free market evolution, but by centralized planning institutions. Modern Russia stands on the edge of a precipice of bad historic decisions and questionable planning choices (Bond et al. 1990). Decades of centralized development left Russia with economic realities which are not translatable to the free market system of today. Russia can be described not as an underdeveloped country, but as a badly developed country, the economic system of which is teeming with various problems of archaic heritage which cause regional misbalances. Similar explanations can be found among mainstream academic ideas, such as Dutch disease and natural resource theories. Additionally, as far as we can tell, among multitude of countries, these problems are especially prevalent among post-Soviet states (Sadik-Zada et al. 2021; Elkhan Richard 2021; Niftiyev 2020; Lovec and Juvančič 2021).

This study aims to develop a model that makes it possible to determine the AEM value, considering the investment accelerator action based on the self-similarity of the multiplier processes.

The research question is whether autonomous expenditure values (AEM) can be used as a tool for macroeconomic policymaking and whether AEM values can represent real world economic processes.

Hypothesis can be formulated as follows: if autonomous expenditure values can be used in macroeconomic policymaking, then it should be possible to formulate model which allows to account for regional specificity.

The model should represent real economic situation processes and values. To prove the model's efficacy, we shall establish AEM values for the economies of the five regions of the Central Federal District using the new model.

The work is presented in the form of structural sections. The Section 1 discusses and substantiates the necessity of applying AEM and defines the study's hypothesis and objectives. The Section 2 contains the analysis of the main metrics of regional ranking in terms of their investment and social attractiveness; at the same time, it presents the investment structure of the Russian Federation and the literature on existing methods of investment management. The Section 3 section reveals the author's approach to the AEM further disclosed in Results using the example of the Central Federal District regions' analysis. The Section 4 makes it possible to see whether the obtained results correspond to the study hypotheses, and the Section 5 briefly presents the author's results obtained during the study.

## 2. Literature Review

From an applied point, one of the main methods for determining the investment attractiveness and feasibility is integrated rating indicators of government bodies and expert companies.

The international practice of regional and municipal ranking is conditioned by national peculiarities of territorial and social development of one or another country. The American models of the USA assessment are of particular interest. Throughout its history, regional independence in the USA has been characterized by a high level of autonomy, which is preserved to this day. The USA has inherently unique tools and methods for balancing the Federal government's capabilities against the constitutional rights of individual states. This structure inevitably imposes its specifics on how states' long-term investment opportunities are assessed(U.S. News & World Report 2021).

The main indicators are the methodologies of major rating agencies such as S&P Global, Moody, and Fitch. Each of these agencies assesses the solvability of states on government loans and ranks them according to their reliability (S&P Global Ratings 2021). In terms of the investment attractiveness of individual states, we can also highlight the most extensive rating systems and models conducted by the major media agencies. "US news & world report state ranking" (2021) is the most extensive ranking survey, aiming to show the most competitive and socially oriented states. In calculating the rankings, each of the eight categories is assigned weights based on annual nationwide surveys in which nearly 70,000 people prioritize each category in their state. Based on this, a metric index score is created for each state, with the relationships between them indexed proportionally. After converting the raw data into index scores for each state, they are averaged to determine scores and rank subcategories.

As for the rating and ranking of Russian regions, it is worth noting that the ratings of international organizations are practically absent. The Russian system of independent evaluation of regions is characterized by high self-sufficiency. Within the practice of strategic investment planning in Russia, the following ratings of regions can be distinguished (Table 1).

**Table 1.** Russian ratings of regions.

| Rating | Developer | Brief Description | Rating Leaders |
|---|---|---|---|
| Rating of the socio-economic situation of the RF regions (RIA Rating 2021a) | RIA Rating | Rating is compiled based on 18 indicators, ranked into 4 groups. The indicators are characterized by their high level of reliability. The main data source is the official statistical compilations of the Ministry of Finance, Treasury, and Rosstat. It provides an overall assessment of the regions. | Moscow St. Petersburg Khanty-Mansi Autonomous Okrug–Ugra |
| Quality of life rating in Russian regions (RIA Rating 2021b) | RIA Rating | Rating is compiled based on 72 indicators, ranked into 11 groups. The indicators are characterized by their high level of reliability. The main data source is the official statistical compilations of the Ministry of Finance, Treasury, and Rosstat. | Moscow St. Petersburg Moscow region |
| Credit rating of Russian regions (RA Expert 2021) | RIA Rating | It is an aggregated indicator that characterizes the ability of the region to service targeted government loans and credits formed within three groups of indicators (budget, debt load, and economy). The index is formed using a scale from 0 to 100 based on 14 statistical indicators. | Tatarstan Bashkortostan Irkutsk region |

**Table 1.** *Cont.*

| Rating | Developer | Brief Description | Rating Leaders |
|---|---|---|---|
| Debt level rating of Russian regions (RIA Rating 2021c) | RIA Rating | Rating characterizes the overall level of debt load in the context of the total share of government debt. | Sevastopol Moscow Tyumen region |
| Scientific and technological development rating of Russian regions (RIA Rating 2020) | AV Group | It is a composite index calculated based on 19 indicators ranked in 4 groups. | Moscow Tatarstan St. Petersburg |
| Competitiveness index of Russian regions AV RCI (Leontiev Centre and AV Group 2020) | RAEX | It is an integrated indicator showing the overall ability of regions to compete for resources and markets formed based on a large amount of public data, statistical indicators, and expert opinions. The obtained indicators are divided into 7 groups of development factors, from which a scale from 0 to 5 is derived, where 2.5 is an average indicator for Russia. | Moscow St. Petersburg Moscow region |
| Investment attractiveness rating of Russian regions (RAEX Analytic 2021) | RAEX | It is the oldest rating compiled since 1996. Over the past 25 years, the methodology of calculating the rating has not changed at all, and it is calculated based on 9 particular potentials (labor, production, and others). Regions are divided into 13 groups, each of which characterizes the relationship between investment potential and risk. | Moscow region Moscow St. Petersburg |
| National investment rating in the RF regions (Agency for Strategic Initiatives 2021) | ASI | Based on 44 indicators, ranked by 5 areas, a special questionnaire is formed, which is used to compile a rating of each region with the help of expert opinions. | Moscow Tatarstan Tyumen region |
| Assessment of investment attractiveness of Russian regions (National Rating Agency 2021) | NRA | Calculated indicators and their basic and critical values are compiled using 56 indicators. Subsequently, based on the expert weights, the aggregate evaluations of the factors and the composite index of each region are calculated. The subsequent cluster analysis makes it possible to formulate the distribution of regions into three indicators and nine groups, each of which represents a certain level of investment attractiveness. | Moscow St. Petersburg Yamalo-Nenets Autonomous Okrug |

However, despite a self-sufficient and independent system of region ratings, their utilitarian connection with the actual processes of investment activity is rather weak.

For example, looking at the investment structure in fixed capital, we can observe a very high level of investment concentration in the key central districts and federal cities. At the same time, the rest of the country is characterized as an investment periphery (Table 2).

Table 2 shows that the combined share of Moscow, St. Petersburg, Moscow Region, and Leningrad Region in the total investment in fixed capital in the Russian economy in 2018 was 26.6%. At the same time, the share of Moscow is 14.05%, comparable to all investments in fixed capital in the economy of the Volga Federal District.

Analyzing the regional structure of investment in fixed capital in dynamics, we can conclude that the regional polarization of investment increased after 2014—both a cause and a consequence of a more developed economy and infrastructure in the central regions of Russia.

**Table 2.** Investment structure of the Russian Federation in the distribution by federal districts (Federal State Statistics Service 2021a).

| Region | 2000 | 2006 | 2009 | 2011 | 2012 | 2015 | 2016 | 2017 | 2018 | 2019 | 2020 |
|---|---|---|---|---|---|---|---|---|---|---|---|
| Central Federal DistrictwithoutMoscow and Moscow region | 8.32 | 8.42 | 10.1 | 10.44 | 9.73 | 10.2 | 9.91 | 9.57 | 8.94 | 8.97 | 8.23 |
| Moscow | 13.41 | 12.5 | 9.31 | 7.76 | 9.69 | 11.1 | 11.6 | 12.5 | 14.1 | 16.91 | 17.73 |
| Moscow region | 4.35 | 5.01 | 4.77 | 4.07 | 4.11 | 4.49 | 4.21 | 4.37 | 5.19 | 5.64 | 5.23 |
| Northwestern Federal Districtwithout St. Petersburgand Leningrad region | 5.28 | 6.98 | 5.12 | 6.02 | 6.38 | 5.24 | 5.42 | 5.45 | 5.06 | 4.74 | 4.81 |
| St. Petersburg | 3.08 | 4.09 | 4.19 | 3.27 | 2.8 | 3.48 | 4.6 | 4.2 | 4.32 | 3.84 | 3.86 |
| Leningrad region | 1.65 | 2.69 | 2.39 | 2.77 | 2.63 | 1.63 | 1.79 | 2.11 | 2.7 | 2.21 | 2.27 |
| Southern Federal District | 9.35 | 6.86 | 8.89 | 9.78 | 9.97 | 9.33 | 7.82 | 9.01 | 8.13 | 7.12 | 7.21 |
| North Caucasian Federal District | 2.23 | 2.73 | 3.35 | 3.15 | 3.2 | 3.42 | 3.3 | 3.1 | 3.19 | 3.25 | 3.51 |
| Volga Federal District | 17.75 | 16.6 | 16 | 15.43 | 16 | 17.7 | 16.5 | 15.2 | 14.3 | 14.06 | 13.73 |
| Urals Federal District | 21.52 | 16.9 | 16.8 | 16.66 | 16.2 | 17 | 18.2 | 17.7 | 17 | 15.35 | 15.64 |
| Siberian Federal District | 7.58 | 9.42 | 9.64 | 10.21 | 10.7 | 9.14 | 8.99 | 8.82 | 9.11 | 9.3 | 9.51 |
| Far Eastern Federal District | 5.48 | 7.8 | 9.43 | 10.45 | 8.58 | 7.32 | 7.59 | 8.04 | 8.07 | 8.59 | 7.71 |
| Moscow St. Petersburg, Moscow, and Leningrad region | 22.49 | 24.3 | 20.7 | 17.87 | 19.2 | 20.7 | 22.2 | 23.2 | 26.3 | 28.6 | 29.09 |

Thus, the economy of the Russian Federation, being initially in unfavorable economic conditions, has recently faced a new combination of negative social, political, and economic factors, from actions of which the regions, remote from the center of the country, suffer to a greater extent. The rapid and large-scale dynamics of the crisis, affecting all sectors of the economy, the uncertainty of the exit timing from this crisis makes it unlike others.

The situation is aggravated by the decline in federal and regional budget revenues. Thus, the effect of the COVID-19 pandemic on the Russian economy was the reduction in tax and non-tax revenues of the budgets of Russian regions by 5.2% in the first nine months of 2020, compared with the same period in 2019. Total revenues of consolidated regional budgets increased but at the expense of increasing the number of transfers from the federal budget by 57.3% (Federal State Statistics Service 2021b).

At the same time, the federal budget revenues for 9 months in 2020 decreased by 1.9 pp of GDP compared with the same period in 2019 due to a reduction in oil and gas revenues of the federal budget. Simultaneously, the federal budget expenditures in 9 months in 2020 have increased by 4.2 pp. One of the results of such dynamics of the federal budget revenues and expenditures was that domestic borrowing in 9 months in 2020 exceeded by 1.2 times borrowing approved for 12 months in 2020 (Federal State Statistics Service 2021b).

An interesting point can be identified in reviewing the latest COVID-19 government spending research (Derkacz 2020, 2021a, 2021b), as it shows similar changes in consumer behavior and government spending. Of interest can also be studies that establish the connection of government spending at different time frames (Quaas 2015), as well as the overall influence of such spending (Bayer et al. 2020).

In our opinion, overcoming the regional polarization of investments is one of the most effective responses to negative factors. An impetus to the economic development of Russian regions is possible as a result of the implementation (especially in conditions of falling incomes) of investment projects that create jobs, increase tax revenues of the regions and the center, incomes of the population, thus counteracting the concentration of the country residents in its central part.

However, reducing available investment resources (private and government) and the increasing requirements to the efficiency of their application set new requirements to the management mechanism of interregional distribution of investments. More and more often, this mechanism faces the task of selecting one point of investment application from many

possible ones, and, in particular, the task of selecting those points on the regional-industrial map of Russia, where implementing investment projects will help obtain significant rates of economic growth, spreading to neighboring regions and industries. The task of creating such points is equally important.

The existing studies of the multiplier largely consist in the calculation of its values based on matrix and various regression models and do not imply the multiplier use as a tool for interregional distribution of investment (Arogundade et al. 2021). Russian researchers use regression models to prove the hypothesis of the positive multiplier effect of investment in fixed capital on the dynamics of Russian GDP (Nikolaev et al. 2019), conducting comparative modeling for different periods (Grabova and Grabov 2019). They calculate the multiplier value for the economies of particular regions of Russia, for example, Tatarstan (Gorid'ko and Nizhegorodcev 2018). Industry multiplier values for the Russian economy are calculated using the input-output matrix (Ksenofontov et al. 2018) for comparative analysis of investment multiplier values for the Russian and U.S. economies (Suvorov 2014).

Foreign authors also pay close attention to the study of multiplier effects, analyzing the effectiveness of government spending as an impetus for developing the country's crisis economy, including the contribution of the investment multiplier mechanism in spreading and strengthening this impetus (Cohen et al. 2011), (Chodorow-Reich et al. 2012).

Within the use of input-output balance to assess values of multiplier effects, foreign researchers use three main approaches:

1. General models of economic equilibrium with the integration of input-output tables (Burfisher 2017).
2. Static models of input-output balance (Miller and Blair 2009).
3. Input-output balance models modified by econometric models (Ghosh et al. 2011).

In addition, the literature also widely presents ideas and studies of fiscal policy dependence, consumer behavior, and their impact on the AEM (Zhang et al. 2020). The interrelation of the studied parameters during crisis periods (Kameda et al. 2021) and the influence and role of public procurement on firms' capital investments in a difficult financial situation are gaining more and more applied significance (Fritsche et al. 2021).

At the same time, it is worth noting the studies which argue that there is no relationship between secondary factors capable of having a long-term impact on investment potential and reducing real investment to short-term government expenditures (Chodorow-Reich et al. 2012). Some researchers also pay special attention to government spending in terms of military capital expenditures and their connection with the investment multiplier (Ramey and Zubairy 2018).

The research results obtained indicate the low values of the government spending multiplier, both for the Russian economy (Drobyshevsky and Nazarov 2013) and for foreign economies (Barro and Redlick 2011), confirming the need for the point application of government investment to use for the economic development, not the multiplier of government spending with its low values but the multipliers of those industries and regions, where investments will be purposefully directed.

The question put forward in this research is whether autonomous expenditure values (AEM) can be used as a tool for macroeconomic policymaking and whether AEM values can represent real world economic processes. It is important to address the research question from a position of its practical application regarding regional investment distribution and its economic feedback. It is essential to approbate the proposed model by means of empirical examination. As an example, the authors chose closely related and deeply economically codependent regions of Moscow, the Moscow region, Kaluga region, Yaroslavl region, and the Ryazan region.

The dynamics of budget revenues and expenditures described above, along with the scale and dynamism of the COVID crisis development, makes it difficult to use traditional tools for managing the Russian economy dynamics, such as instruments of monetary and

fiscal policy. The increased expenditures and decreasing revenues of the budgets require a more thoughtful and reasonable application of these tools.

## 3. Materials and Methods

The materials applied in this study are statistical data of the Central Federal District regions in the furtherance of the tasks. At the same time, it is worth noting that the statistical sample was determined by the factors of the Russian Federation statistical system with its practice of retrospective revision and methodological transformation of already prepared statistical compilations.

The work applies scientific methods of comparative analysis, deduction, and mathematical modeling.

The methodology of this study can be presented in the Figure 1:

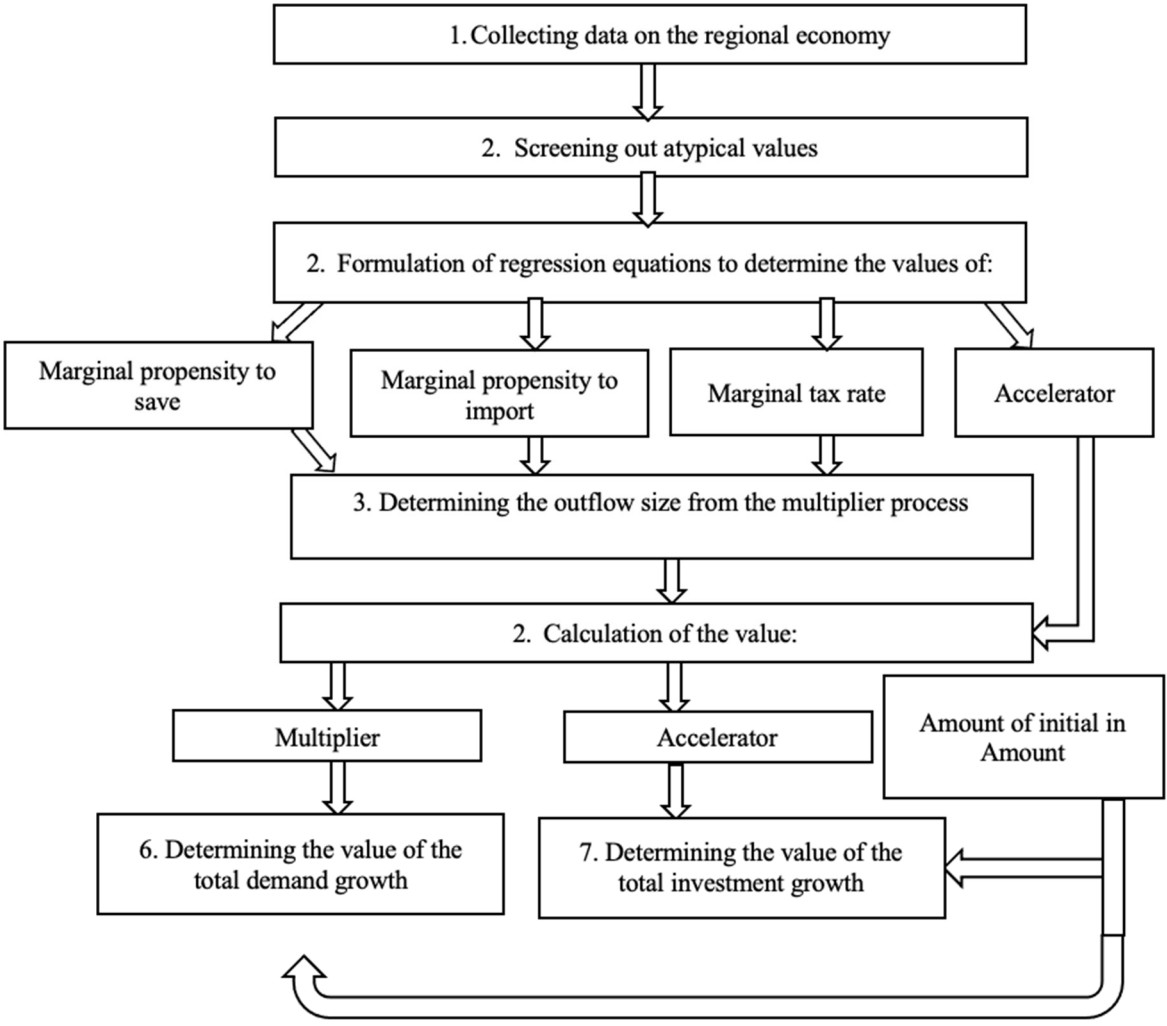

**Figure 1.** Research design.

Research design establishes and outlines necessary steps in framing and proving validity of the model we propose. First step in establishing working model is data aggregation and data screening, which allow us to formulate regression equations aimed at determining "marginal" parts of the model as well as accelerator values. Based on these values we can determine multiplier outflows. Final steps consist of calculating multiplier and accelerator values which in turn allow us to calculate values of total demand and investment growth.

The analysis of works by Russian and foreign authors allowed us to conclude that they are mainly devoted to calculating and analyzing the investment multiplier or AEM values

and do not suggest its use as an active tool for managing the development of the country's economy. The use of paired linear regression models to model multiplier processes does not make it possible to describe the internal dynamics of these processes necessary to manage them. The use of matrix models is a more promising way because it allows us to determine the sectoral distribution of multiplier effects. However, since the development of such models within the country is rather labor-intensive and carried out with significant time delays, their use to solve the practical problems of investment allocation management seems difficult.

The above-described multiplicative chains and the investment accelerator action supporting their development are formed of different-scale recurring components that allow us to model the interaction between AEM and the investment accelerator as a self-similar, fractal process. The fractal model of the multiplier–accelerator, supplemented with econometric models, is proposed to manage the interregional investment distribution. Let us consider it in more detail by formalizing the first and subsequent multiplicative chains in expression (1).

$$AD_1 = IN_1 + IN_1 \times (1 - MO) + IN_1 \times (1 - MO)^2 + \ldots + IN_1 \times (1 - MO)^n \quad (1)$$

where

$AD_1$ is the aggregate demand imposed by resource suppliers–participants of the first multiplicative chain;
$IN_1$ is the autonomous investment amount made by the first investor;
$IN_1 \times (1 - MO)$ isthe demand volume of the *i*-th resource supplier, taking into account the outflows from this process;
MO is the marginal value of outflows from the multiplicative chain.

Here

$$MO = MPS + MPI + MPT \quad (2)$$

where

*MPS*–the value of the marginal propensity to save ($0 < MPS < 1$)
*MPI*–the value of the marginal propensity to import ($0 \leq MPI \leq 1$)
*MPT*–marginaltaxrate

Savings are outflows from the multiplicative chain, but their investments initiate new multiplicative chains in other multiplier processes (Suvorov 2014). The situation is similar to tax payments. Taxes to the budgets of all levels are outflows from a particular multiplicative chain, weakening it (the marginal tax rate characterizes the value of these outflows). However, tax-financed government spending is the initial investment impulse generating new chains that increase the aggregate volume of the multiplier effect. Import expenditures transfer the multiplicative chain outside the region under study, directing its effect to develop the economies of other regions (countries). Consequently, the marginal propensity to import characterizes the outflow that weakens the multiplicative chain. However, exports and foreign spending on Russia's products and services are the initial investment impulses forming new multiplicative chains determining the division of net exports with the attribution of imports to outflows and exports – to multiplier impulses.

For determining the values of *MPS*, *MPI*, *MPT*, it is proposed to complement the described model of interaction between the AEM and the investment accelerator with paired linear regression models of the form:

$$y = a + b \times x \quad (3)$$

where

*x* is the volume of gross regional product (GRP) for the analyzed period;
the values of *a*, *b*, and *y* are in the following dependence:

- If *b* is (1-*MPS*), then *y* is the volume of consumer spending in the economy of the region in the analyzed period, and *a* is the volume of autonomous consumer spending;
- If *b* is *MPI*, then *y* is the volume of imports into the regional economy in the analyzed period, and *a* is the volume of autonomous import expenditures;
- If *b* is *MPT*, then *y* is the volume of tax payments in the region's economy in the analyzed period, and *a* is the volume of accord tax payments.

It should be emphasized that this paper proposes using AEM as a management tool for the interregional distribution of investment. Then, to calculate its components, it is proposed to use the GRP volume, formed not only by autonomous but also by induced expenditures. The reason for this is that the multiplier action mechanism, which consists of the formation of income transmission chains, supported by additional investment impulses of the accelerator, is the same for both the autonomous and induced spending multiplier, as well as the aggregate spending multiplier since the set of participants in multiplicative chains for any of these multipliers is the same.

First of all, it is many individuals (employees and business owners) and groups of resource and equipment suppliers. The same composition of participant groups, which scales erase individual differences in the volume of outflows from the multiplier process, brings together the values of investment multipliers, autonomous and aggregate costs within the same industry (region) to the degree of identity. The approximation process is also facilitated by the fact that both autonomous expenditures and aggregate expenditures include autonomous investments, and also that even the initial autonomous expenditures for particular resource suppliers become additional revenues, leading to an increase in their consumption, thus including induced consumption in the AEM chain at some stages.

Therefore, within one branch or, given the branch structure of the economy, one region, the multiplier values of autonomous and aggregate expenditures, will be brought together to the degree of identity. Extremely insignificant differences will exist, but at this stage of development of the proposed tool, they can be neglected, returning to their analysis with sufficient fine-tuning of the proposed tool based on the collected practical experience of its application.

The identity of the multiplier values of autonomous and aggregate expenditures makes it possible to use the AEM value of GDP (GRP) for calculating.

At the same time, there are differences between the multipliers of autonomous and aggregate expenditures. They lie in the volume of the initial investment impulse and, consequently, the total volume of the multiplier effect. Thus, the initial AEM impulse is autonomous spending (with one volume). In contrast, the initial impulse of the aggregate spending multiplier is aggregate spending larger in volume than autonomous spending.

Formula (1) is simplified by transforming it into the product of the autonomous investment volume of the first investor by the coefficient of simple (without investment accelerator) AEM:

$$AD_1 = \lim_{n \to \infty} \sum_{i=0}^{n} IN_1 \times (1 - MO)^i = IN_1 \times \frac{1}{MO} \tag{4}$$

Let us add the investment accelerator action to the analysis.

$$IN_2 = IN_1 \times (1 - MO) \times A + IN_1 \times (1 - MO)^2 \times A + \ldots + IN_1 \times (1 - MO)^n \times A \tag{5}$$

where

$IN_2$ is the amount of investment made by investors to meet demand in the volume of AD1;
A is the value of investment accelerator in the analyzed economy ($0 \leq A \leq 1$), it is economically unprofitable to invest if A > 1; in the case of excess of free production capacity A = 0;
$IN_1 \times (1 - MO)^i \times A$ is the amount of investment to meet the demand of the *i*-th resource supplier.

The value of investment accelerator A is proposed to determine by an econometric model of the form (3) where:

*x* is the GRP volume for the analyzed period;
*a* is the volume of autonomous investments in the regional economy during the analyzed period;
*b* is the required value of investment accelerator;
*y* is the volume of investments in the region's economy during the analyzed period.

As a result of simplification, Formula (5) takes the form (6):

$$\text{IN}_2 = \lim_{n \to \infty} \sum_{i=0}^{n} \text{IN}_1 \times (1 - \text{MO})^i \times A - \text{IN}_1 \times A = \text{IN}_1 \times \frac{(1 - \text{MO}) \times A}{\text{MO}} \tag{6}$$

It can be seen that Formulas (1) and (5) and, consequently, Formulas (4) and (6), consist of recurring elements of different scales, which allows us to simplify the following steps in modeling the process of interaction between the AEM and the investment accelerator.

The investment volume IN2 generates its multiplicative chain. The demand volume of its participants (AD2), according to Formula (4), is calculated as:

$$\text{AD}_2 = \text{IN}_2 \times \frac{1}{\text{MO}} = \left[ \lim_{n \to \infty} \sum_{i=0}^{n} \text{IN}_1 \times (1 - \text{MO})^i \times A - \text{IN}_1 \times A \right] \times \frac{1}{A} = \text{IN}_1 \times \frac{(1 - \text{MO}) \times A}{\text{MO}^2} \tag{7}$$

Similar to the AD1 volume, the AD2 volume will also require the similar amount of investment (IN3) to create additional production capacity:

$$\text{IN}_3 = \text{IN}_2 \times \frac{(1 - \text{MO}) \times A}{\text{MO}} = \text{IN}_1 \times \frac{(1 - \text{MO}) \times A}{\text{MO}} \times \frac{(1 - \text{MO}) \times A}{\text{MO}} = \text{IN}_1 \times \left[ \frac{(1 - \text{MO}) \times A}{\text{MO}} \right]^2 \tag{8}$$

The analysis of Formulas (1)–(8) allows us to conclude that the subsequent stages of the analyzed process will develop similarly to the previous ones; therefore, they do not require separate modeling. Based on these formulas and the identified pattern, let us define AD—the total volume of demand formed in the process of interaction between AEM and the investment accelerator as a result of the initial investor (IN1) actions:

$$\text{AD} = \frac{\text{IN}_1}{\text{MO}} + \frac{\text{IN}_1}{\text{MO}} \times \frac{(1 - \text{MO}) \times A}{\text{MO}} + \frac{\text{IN}_1}{\text{MO}} \times \left( \frac{(1 - \text{MO}) \times A}{\text{MO}} \right)^2 + \ldots + \frac{\text{IN}_1}{\text{MO}} \times \left( \frac{(1 - \text{MO}) \times A}{\text{MO}} \right)^n \tag{9}$$

Let us simplify Formula (9) by reducing it to the form (10):

$$\text{AD} = \lim_{n \to \infty} \sum_{i=0}^{n} \frac{\text{IN}_1}{\text{MO}} \times \left( \frac{(1 - \text{MO}) \times A}{\text{MO}} \right)^i = \text{IN}_1 \times \frac{1}{\left( 1 - \frac{(1-\text{MO}) \times A}{\text{MO}} \right) \times \text{MO}} \tag{10}$$

Coefficient $M = \frac{1}{\left( 1 - \frac{(1-\text{MO}) \times A}{\text{MO}} \right) \times \text{MO}}$ in Formula (10) isthe AEM coefficient, taking into account the interaction of multiplier and accelerator processes.

Let us similarly determine IN—the total amount of investment formed in the process of interaction between the AEM and the investment accelerator as a result of the initial investor ($\text{IN}_1$) actions:

$$\text{IN} = \text{IN}_1 + \text{IN}_1 \times \frac{(1 - \text{MO}) \times A}{\text{MO}} + \text{IN}_1 \times \left( \frac{(1 - \text{MO}) \times A}{\text{MO}} \right)^2 + \ldots + \text{IN}_1 \times \left( \frac{(1 - \text{MO}) \times A}{\text{MO}} \right)^n \tag{11}$$

Let us simplify Formula (11) by reducing it to the form (12):

$$\text{IN} = \lim_{n \to \infty} \sum_{i=0}^{n} \text{IN}_1 \times \left( \frac{(1 - \text{MO}) \times A}{\text{MO}} \right)^i = \text{IN}_1 \times \frac{1}{1 - \frac{(1-\text{MO}) \times A}{\text{MO}}} \tag{12}$$

where the coefficient before $IN_1$ is the investment accelerator, taking into account the interaction of multiplier and accelerator processes.

The multiplier and accelerator chains contain income and expenses. The application of two Formulas (9) and (11) makes it possible to distinguish them. Thus, the multiplier chain (Formula (9)) describes the income transfer to its different participants. At the same time, expenditures are only an intermediary, the attention on which is not focused. The accelerator chain (Formula (11)) describes the emergence of additional expenses as additional investment impulses, allowing the multiplicative chain to avoid premature fading.

Using the multiplier coefficient from Formula (10), let us calculate the values of AEM for the regions of the Central Federal District, taking into account the investment accelerator action, proving not only the applicability of this formula in practice but also the difference in the values of regional AEMs, which is the basis for applying the multiplier as a tool for interregional distribution of investment.

The practical application of the multiplier formula from Expression (10) faces some difficulties in the calculation information support. In order to consider the regions as closed systems, it is necessary, such as with the states, to calculate the volume of their imports and exports for the economies of these regions. Although for the economies of particular states, this task is solved quite simply based on the customs data analysis (smuggling complicates this process), for a particular region, the volumes of exports and imports are calculated as follows:

$$Im_r = Im_{in} + Im_{or} \tag{13}$$

$$Ex_r = Ex_{in} + Ex_{or} \tag{14}$$

where

$Im_r(Ex_r)$ is the volume of regional imports (exports);

$Im_{in}$ ($Ex_{in}$) is the volume of imports of foreign products (exports of regional products outside the country) in the economy of the analyzed region;

$Im_{or}$ ($Ex_{or}$) is the volume of imports (exports) of products from other regions of the country (to other regions) in the economy of the analyzed region.

The creation of special services tracking the volume of regional exports and imports is rather costly, which, especially in the field of regional imports, will not bring the good effect since it is extremely difficult to track the volume of imports into the region's economy because of an extremely large number of points beyond its borders.

Let us combine Formulas (13) and (14) by expressing the volume of regional imports via the volume of regional exports, using the formula for determining the GRP volume (15) to solve this problem.

$$GRP = C_r + I_r + G_r + EX_r - IM_r \tag{15}$$

where

$GRP$ is the gross regional product volume for a given period;

$C_r, I_r, G_r$ are the volumes of regional consumer spending, gross private domestic investment, and government spending, respectively, for a given period.

From Formula (15), it follows:

$$IM_r = C_r + I_r + G_r - (GRP - EX_r) \tag{16}$$

The values of all economic indicators from Equation (16) right side, except $EX_r$, can be obtained from publicly available regional statistics data

Information on $EX$ volumes is available only for the more developed regions of the central part of the Russian Federation. For the regions of Siberia and the Far East, this information is not available.

However, obtaining this information within a current digital economy is not difficult. In order to solve this problem without additional costs and resistance from enterprises, it is proposed to oblige several Russian companies that supply accounting automation

software to update it so that, based on primary accounting documents containing buyers' addresses, this software would form a monthly combined statement containing the volume of products in monetary terms, sold by an enterprise outside the region.

## 4. Results

Let us calculate the AEM value for Moscow and the regions surrounding Moscow: Moscow, Kaluga, Yaroslavl, and Ryazan regions. Initial data for the calculation are presented selectively in Table 3. The data are presented selectively since their initial set is extremely large both by region and by time. Table 3 establishes forward movement of the economy yet highlight certain distinct differences between regions especially Moscow, which stands by itself in almost every value, some of these values overshadow its neighbors by a tenfold. As a capital of Russia, Moscow enjoys privileges of housing headquarters of major corporations, interest groups, and political parties. In certain sense, Moscow can be compared with New York, Washington DC, and Los Angeles, but in one place. As such, it significantly skews any comparative research, yet it is important to add it as a comparison as it highlights any model usefulness with its high values of an absolute economic outlier.

**Table 3.** Initial data (selectively) to calculate regional AEM values (Federal State Statistics Service 2021b).

| Indicator | 2005 | 2011 | 2012 | 2014 | 2017 |
|---|---|---|---|---|---|
| Moscow region | | | | | |
| Population, thousand people | 6783.8 | 7198.7 | 7048.1 | 7231.1 | 7503.4 |
| GRP, mln. RUB | 708,062 | 2,176,795 | 2,357,082 | 2,742,886 | 3,802,953 |
| Investment in fixed capital, mln. RUB | 181,260 | 449,666 | 516,872 | 644,830 | 699,918 |
| Consumer spending per capita (per month), RUB | 6077 | 18,209 | 20,553 | 25,576 | 32,159 |
| Tax revenues (fees) to the budget system of the Russian Federation, mln. RUB | 157,666 | 444,374 | 510,452 | 600,202 | 832,515 |
| Value of regional import, mln. RUB | 92,150 | 243,321 | 250,152 | 295,591 | 422,198 |
| Moscow | | | | | |
| Population, thousand people | 10,923.8 | 11,612.9 | 11,979.5 | 12,197.6 | 12,506.5 |
| GRP, mln. RUB | 4,135,155 | 9,948,773 | 10,666,871 | 12,779,526 | 15,724,910 |
| Investment in fixed capital, mln. RUB | 456,025 | 856,424 | 1,220,097 | 1,541,884 | 2,007,708 |
| Consumer spending per capita (per month), RUB | 16,961 | 34,585 | 37,488 | 47,966 | 51,069 |
| Tax revenues (fees) to the budget system of the Russian Federation, mln. RUB | 801,856 | 2,038,366 | 2,166,699 | 2,233,836 | 3,068,726 |
| Value of regional import, mln. RUB | 376,269 | 889,425 | 951,124 | 1,094,263 | 1,379,312 |
| Kaluga Region | | | | | |
| Population, thousand people | 1023.3 | 1008.2 | 1005.6 | 1010.5 | 1012.2 |
| GRP, mln. RUB | 70,954 | 234,749 | 285,257 | 326,459 | 417,065 |
| Investment in fixed capital, mln. RUB | 13,624 | 77,354 | 95,970 | 99,786 | 89,030 |
| Consumer spending per capita (per month), RUB | 4129 | 12,886 | 14,525 | 19,029 | 21,892 |
| Tax revenues (fees) to the budget system of the Russian Federation, mln. RUB | 14,501 | 46,068 | 63,758 | 73,699 | 92,060 |
| Value of regional import, mln. RUB | 8903 | 25,296 | 29,347 | 33,467 | 42,528 |

Three paired linear regression models for each of the analyzed regions were formed based on the initial data. Fifteen paired linear regression models were made. Let us consider an example of regression models obtained for the economy of the Moscow region:

$$C = (1 - MPS) = (1 - 0.2507) \times GRP \tag{17}$$

$$I = A \times GRP = 0.2082 \times GRP \tag{18}$$

$$T = MPT \times GRP = 0.2120 \times GRP \tag{19}$$

$$IM = MPI \times GRP = 0.1041 \times GRP \tag{20}$$

where

*GRP* is the gross regional product value in the regional economy;
*C* is the consumer spending value in the regional economy;
*I* is the investment in fixed capital value in the regional economy;
*T* is values of tax revenues, fees, and other compulsory payments from the regional economy to the budgets at all levels of the budget system of the Russian Federation;
*IM* is the value of regional import.

Note that the accelerator is understood as the investment increment in the current period at the expense of costs of the previous period. In this case, investment (I) and GRP in Formula (18) refer to the same period due to the scale of the analyzed period. If investments resulting from earnings gain were not made during the year or quarter and will be made only in the next year (quarter), the difference between GRP and investments in Formula (18) should be one period. However, this approach describes the investment response to earnings gain as somewhat discrete, slowing it down. In practice, the reaction of producers to earnings gain in the form of additional investments can be rather fast and occurs within a month (i.e., within the analyzed period). Such reaction requires a continuous rather than discrete approach to investment analysis, thus conditioning the application of Formula (18) in the presented form.

Coefficients a in paired linear regression equations were not calculated because preliminary modeling showed for some regions their dependence on the number of credits stimulating consumer spending, often without their connection with the income dynamics, and on inter-budget transfers (in determining the marginal tax rate). At the same time, when forming a tool to find the optimal points of investment application to the regional economy, it is necessary to rely on the multiplier effect formed by real economic activity, rather than "inflated" by consumer lending or the movement of budgetary funds.

The corresponding values of the coefficients for other analyzed regions, according to notations adopted in Formula (2), and estimates of statistical significance of the generated paired linear regression equations and their parameters are presented in Table 4.

**Table 4.** Coefficients and importance factors of paired linear regression equations.

| Region | Coefficient *b* | Value of *b* | Importance of Paired Linear Regressionequations | | | | | |
|---|---|---|---|---|---|---|---|---|
| | | | *R*-Squared | Norm. *R*-Squared | Importance *F* | Standard Error *b* | *t*-Statistics | *p*-Value |
| Moscow | *MPS* | 0.4937 | 0.9954 | 0.9121 | $2 \times 10^{-14}$ | 0.0099 | 51.0878 | $2 \times 10^{-15}$ |
| | A | 0.1151 | 0.9897 | 0.9064 | $1.7 \times 10^{-12}$ | 0.0034 | 34.0048 | $3 \times 10^{-13}$ |
| | *MPT* | 0.1949 | 0.9873 | 0.9039 | $5.6 \times 10^{-12}$ | 0.0064 | 30.4872 | $1 \times 10^{-12}$ |
| | *MPI* | 0.0874 | 0.9902 | 0.9893 | $2.11 \times 10^{-12}$ | 0.0026 | 33.3422 | $2 \times 10^{-11}$ |
| Moscow region | *MPS* | 0.2507 | 0.9981 | 0.9148 | $1.4 \times 10^{-16}$ | 0.0093 | 80.3799 | $9 \times 10^{-18}$ |
| | A | 0.2082 | 0.9747 | 0.8913 | $2.5 \times 10^{-10}$ | 0.0097 | 21.481 | $6 \times 10^{-11}$ |
| | *MPT* | 0.2102 | 0.9987 | 0.9154 | $2 \times 10^{-17}$ | 0.0022 | 95.682 | $1 \times 10^{-18}$ |
| | *MPI* | 0.1041 | 0.9852 | 0.9838 | $2.02 \times 10^{-11}$ | 0.0038 | 27.0958 | $2 \times 10^{-13}$ |
| Kaluga region | *MPS* | 0.3202 | 0.9972 | 0.9139 | $1.3 \times 10^{-15}$ | 0.0104 | 65.4403 | $1 \times 10^{-16}$ |
| | A | 0.2856 | 0.9554 | 0.872 | $5.7 \times 10^{-9}$ | 0.0178 | 16.0241 | $2 \times 10^{-9}$ |
| | *MPT* | 0.2082 | 0.9949 | 0.9116 | $3.5 \times 10^{-14}$ | 0.0043 | 48.4881 | $4 \times 10^{-15}$ |
| | *MPI* | 0.0998 | 0.9881 | 0.987 | $6.16 \times 10^{-12}$ | 0.0033 | 30.2205 | $9 \times 10^{-13}$ |
| Yaroslavl region | *MPS* | 0.3856 | 0.9961 | 0.9128 | $8.1 \times 10^{-15}$ | 0.0111 | 55.4387 | $8 \times 10^{-16}$ |
| | A | 0.2143 | 0.9609 | 0.8775 | $2.7 \times 10^{-9}$ | 0.0125 | 17.1648 | $8 \times 10^{-10}$ |
| | *MPT* | 0.2599 | 0.9945 | 0.9112 | $5.2 \times 10^{-14}$ | 0.0056 | 46.79 | $6 \times 10^{-15}$ |
| | *MPI* | 0.0708 | 0.9875 | 0.9864 | $8.1 \times 10^{-12}$ | 0.0024 | 29.4668 | $7 \times 10^{-11}$ |
| Ryazan region | *MPS* | 0.2973 | 0.9976 | 0.9143 | $5.9 \times 10^{-16}$ | 0.01 | 70.4424 | $4 \times 10^{-17}$ |
| | A | 0.2104 | 0.9446 | 0.8613 | $1.9 \times 10^{-8}$ | 0.0147 | 14.3044 | $7 \times 10^{-9}$ |
| | *MPT* | 0.2769 | 0.9892 | 0.9059 | $2.3 \times 10^{-12}$ | 0.0084 | 33.1347 | $4 \times 10^{-13}$ |
| | *MPI* | 0,0631 | 0.9804 | 0.9786 | $9.6 \times 10^{-11}$ | 0.0027 | 23.4496 | $9.8 \times 10^{-12}$ |

Based on the values of the coefficients *b* presented in Table 3, using the formula of the coefficient M in Formula (10), let us calculate the AEM values for the analyzed regions (Table 5) and conduct their analysis.

**Table 5.** AEM values for the economies of Central Russia.

| Region | MO = *MPS + MPT + MPI* | AEM Value |
|---|---|---|
| Moscow | 0.7761 | 1.33282 |
| Moscow region | 0.5649 | 2.10791 |
| Kaluga region | 0.6282 | 1.91562 |
| Yaroslavl region | 0.7163 | 1.52558 |
| Ryazan region | 0.6373 | 1.78241 |

The AEM value in Table 4 is calculated using the above formula:

$$M = \frac{1}{\left(1 - \frac{(1-MO) \times A}{MO}\right) \times MO} \tag{21}$$

The economy of the Kaluga region is characterized by the lowest (except Moscow region) among all the analyzed economies by the value of the outflow from the multiplier process (MO), which determines a rather high AEM value, for example, in comparison with Yaroslavl or Ryazan regions. The reason for that is a higher level of economic development of the analyzed region, whose driver was the automotive production cluster. Thus, in 2019 the Kaluga region ranked first in the Central Federal District and ninth in the Russian Federation regarding industrial production per capita. GRP per capita in Kaluga region is higher than in Yaroslavl or Ryazan regions.

In turn, the AEM value of the Ryazan region is higher than the AEM value of the Yaroslavl region since the difference between the values of outflows from multiplier processes unfolding in these regions, namely, in the higher value of the marginal propensity to save in the Yaroslavl economy.

We should pay special attention to the low (compared with other economies) AEM value in the Moscow economy (which looks illogical, given the scale of the Moscow economy) and, simultaneously, the highest among the calculated AEM values for the Moscow region economy. These values are connected.

The value of the marginal propensity to save (calculated in Table 3 for the economy of Moscow (0.4937)) is overestimated. An important part of Moscow's GRP is created by the labor force coming from nearby regions, especially from the Moscow region. The push-pull migration leads to the fact that these labor resources, receiving incomes in Moscow, spend them mainly on the territory of the Moscow region. As a result, a part of GRP formed in Moscow is spent on the territory of the Moscow region, which underestimates the value of marginal propensity to consume in Moscow economy and overestimates it in the Moscow region. The result is the overestimated MO value for the Moscow city economy, and the underestimated MO value for the Moscow region economy, which, in turn, underestimates the AEM value for the Moscow economy, overestimating it for the Moscow region economy.

## 5. Discussion

The analysis of the works of Russian and foreign authors allows us to conclude that they are mainly devoted to the calculation and analysis of the investment multiplier values (autonomous expenditures) and do not suggest multiplier's use as an active tool for managing the development of the country's economy (Ksenofontov et al. 2018; Grabova and Grabov 2019). The existing studies of the multiplier largely consist of calculating its values based on matrix and regression models and do not imply the multiplier's use as a tool for interregional investment distribution. Through analysis of the literature, we can establish a significant research gap in macroeconomic understanding of how the multiplier can be used as a regional development tool. Earlier, we already compiled a relevant literature review that partially covers the usage of the multiplier as a tool of macroeconomic policy (Gorid'ko and Nizhegorodcev 2018), but in contrast to that study, our results allow us to clearly establish the investment multiplier role in the regional economic development of Russia. Comparing the present study with our previous research (Silvestrov et al. 2018), we

managed to reveal the reserves for the economic development, as well as certain barriers that stand in the way of this development. We have also discovered additional AEM peculiarities of the Moscow region. As our research primarily focuses on Russia, it would be appropriate to note that even among Russia-focused studies, expenditure multiplier research is scarce and is viewed primarily as an economic afterthought, which finds very little utility among academic research (Drobyshevsky and Nazarov 2013).

If we compare existing research with our results, we can observe significant emphasis on how expenditure multipliers can explain connections between secondary factors, such as public debt and income, and very low emphasis on how expenditure multiplier can be used in modeling inter-region connections (Raut and Swati 2019).

As far as regional AEM results go, research is limited by English literature availability; the language barrier creates significant difficulties in data and literature presentation. The majority of studies that focus on AEM, and regional development are written in Russian. As far as comparative analysis can be conducted, our results fall in line with previous research results (Eremin 2020) and models (Eremin 2015).

Present research shows two components that will define future autonomous expenditure multiplier research: data availability and international interest. As of today, relevant literature is constrained not only by language, but also by a lack of similar view in how AEM can be used in managing local investments and development. Broad research interests are limited to state and interstate level (Derkacz 2020). Apart from that, significant interest can be found in how multipliers are affected by coronavirus stimulus spending and whether connections can be established between excessive government spending and diminishing economic activity on the regional scale. Research and data on this topic are rather limited (Butkus et al. 2021), but should improve with time.

Algorithmically, the solution to the problem does not seem to be difficult and costly. Updating the software for businesses that use it in accounting should be free. The current regulatory framework does not require enterprises and organizations to provide information in the proposed form. However, the proposed action to obtain the information does not imply large-scale and revolutionary changes in the regulatory framework. The necessary changes are minimal and can be implemented in the shortest possible time.

## 6. Conclusions

The study shows that multiplicative chains and the investment accelerator action supporting their development form different-scale recurring components that allow us to model the interaction between the AEM and the investment accelerator as a self-similar, fractal process. The AEM application as a management tool for interregional investment distribution will overcome the regional polarization of investment. Interregional investment distribution and subsequent implementation of investment projects will change the values of regional AEMs as a result of changes in the structure of regional economies. The proposed distribution of government investments according to the identified points of their maximum effective application can be carried out legislatively, based on the formed plans and strategies of development. In order to distribute private investments according to the identified optimal regional and sectoral directions, it is necessary to stimulate investors, namely by providing tax benefits, co-financing costs, and developing infrastructure in the places of necessary implementation of investment projects. In addition, the proposed model can be used in assessing the implementation effects of large investment projects, whose implementation is carried out according to the existing policy of special investment contracts, characterized by a high level of flexibility of regional distribution. Results highlighted in Tables 4 and 5 outline both challenges and problems of regional development by means of economic modeling. The study's limitation was the lack of statistical data. The problems were the low level of commercial data availability, which implementation and accounting could significantly improve the accuracy of the proposed model, and the high level of uncertainty of macroeconomic processes caused by the destabilization of international supply and production chains. In the future, it is planned to expand

the AEM model proposed in this work by integrating into it the models that take into account the industry specifics of the regions, investment patterns of technological nature, and that reflect the dependence of the autonomous expenditure models, macroeconomic investment processes, and the dynamics of investment in fixed capital. The proposed approach to investment allocation management based on AEM values can be applied both at the level of state authorities, modeling, planning the economic dynamics and managing its development, and at the level of large corporations, managing their investment program, and forecasting the direct and indirect effects of its implementation.

**Author Contributions:** Conceptualization, S.N.S.; methodology, S.A.P.; software, S.B.R.; validation, S.A.P. and D.V.F.; formal analysis, D.V.F.; investigation, S.B.R.; resources, S.A.P.; data curation, S.N.S.; writing—original draft preparation, S.B.R.; writing—review and editing, D.V.F.; visualization, D.V.F.; supervision, S.A.P.; project administration, S.A.P.; funding acquisition, S.N.S. All authors have read and agreed to the published version of the manuscript.

**Funding:** This research was funded by RFBR according to the research project (grant) No. 19-010-00936.

**Institutional Review Board Statement:** Not applicable.

**Informed Consent Statement:** Not applicable.

**Data Availability Statement:** Information and data related to the subject of the study can be found on the websites of federal executive authorities (in particular, the Ministry of Industry and Trade of the Russian Federation, the Federal State Statistics Service), and various rating agencies.

**Conflicts of Interest:** The authors declare no conflict of interest. The funders had no role in the design of the study; in the collection, analyses, or interpretation of data; in the writing of the manuscript; or in the decision to publish the results.

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
