# Peer review of "Management of the Russian Interregional Investment Distribution Using the Autonomous Expenditure Multiplier Model"

_economies, doi:10.3390/economies10020045_

Round 1
Reviewer 1 Report
I received the developed paper. However, still there is one problem, which I pointed out.
This is related to MPI.
Authors answered as follows.
"In view of the complexity of the study, we have two articles on this topic, the method for
calculating MPI was included in the first article and the only way out would be to quote, but the
imprint of article hasn’t exit in the magazine yet. Therefore, we really ask for help on how to
deal in this case. Thank you!"
However, as we can see MPS and MPT in Table 4 and Equation MO=MPS+MPT+MPI, I cannot check MPI. For example, in Moscow MPS is 0.4937 and MPT is 0.1949, the sum of these is 0.6886, MPI is zero? In all regions, MPI is zero?
Where is MPI?
Author Response
Point 1: I received the developed paper. However, still there is one problem, which I pointed out.
This is related to MPI.
Authors answered as follows.
"In view of the complexity of the study, we have two articles on this topic, the method for
calculating MPI was included in the first article and the only way out would be to quote, but the
imprint of article hasn’t exit in the magazine yet. Therefore, we really ask for help on how to
deal in this case. Thank you!"
However, as we can see MPS and MPT in Table 4 and Equation MO=MPS+MPT+MPI, I cannot check MPI. For example, in Moscow MPS is 0.4937 and MPT is 0.1949, the sum of these is 0.6886, MPI is zero? In all regions, MPI is zero?
Where is MPI?
Responce 1: Our sincere apologies for not being clear on this topic. New document revision contains necessary changes throught the “Materials and Methods” section. We added MPI data and new formulas that account for it addition as well as new final AEM values. Changes can be found at: L410-L412, L439, L445, L473 as well as Table 3, Table 4 and Table 5.
Reviewer 2 Report
I accept changes made by the authors. I recommend this paper to be published in present form.
Author Response
Dear reviewer,
Thank you for the positive verdict.
Reviewer 3 Report
Manuscript title: Management of the Russian Interregional Investment Distribution Using the Autonomous Expenditure Multiplier Model
Overall opinion: I enjoyed reading this well-structured and well-written paper that analyzes the interregional investment distribution using the AEM model and regression estimations. The methodology of the work is sound and reliable. The paper estimates the interregional investment multipliers that serve for better understandings of economic growth based on government and private sector connection on the background of industrial relations. The authors experimented with the collected data that boosts the readability and contribution to the literature. Russia-related literature concerning this theme is scarce, and a research gap needs to be filled. After some major and minor revisions, the work will reach the necessary level of publishability. There are not any significant questionable parts that would reduce the publishability and quality of the research work.
Major comments:
- The abstract section does not include more narrowed and necessary insights about the findings of the paper. The authors argue about the significance and overall importance of the methodology and results; however, precise and accurate positioning of the results is necessary too;
- I strongly recommend to include the end-users of the findings of this paper at the end of the abstract section;
- The citation density of the first two paragraphs of the introduction section is low. Without proper references, introductory sentences are just claims, not argumentation chains. I am sure authors can find proper papers and works to include about overall Russian case;
- Starting from the third paragraph of the introduction section, the authors describe how the regional distribution of the investments is inequal and lags from each other. This situation must be linked to a more general phenomenon in the case of the Russian economy. For instance, Dutch disease and Natural Resource Curse theories widely capture why certain sectors and regions lag due to the booming sectors in a given economy. Therefore, the authors can develop the third paragraph even further by introducing a top-down approach. In other words, the link between the issues of interregional investment distribution and the overall macroeconomic situation might be connected to Dutch Disease. This was found in the other post-soviet countries. Please, refer to Niftiyev (2020) [https://www.econstor.eu/handle/10419/227485], Niftiyev 2021 [https://econpapers.repec.org/article/higecohse/2021_3a2_3a6.htm] and Sadik-Zada (2019) https://link.springer.com/article/10.1007/s13563-019-00202-6 where the Azerbaijani case was analyzed and Kazakhstan https://www.tandfonline.com/doi/abs/10.1080/14631377.2020.1745557 which is similar to the oil and gas-rich Russia;
- L47-49: Some sort of brief literature references should be inserted here to describe and argue about the overall relevance of the applied methodology and research design;
- L59-59: Try to revise the hypothesis. The hypothesis must meet the widely accepted experience of hypothesis-writing. In its current form, it is unsatisfactory;
- L70: I strongly suggest renaming Figure 1 to research design, rather than “methodology” and putting it into the data and materials section. Moreover, Figure 1 has not been explained. The text should reflect the essential ideas of the research design described by the flow chart;
- The literature review section is good, but some minor improvements are needed. Again, similar to the introduction section, some paragraphs of the literature review lack the proper quantity of citations. Moreover, I suggest replacing the paragraph between the lines of L143-148 from its current position, to the end of the section;
- Usually, at the end of the literature review sections, research questions are presented and research objectives are repeated. I strongly suggest doing so which could improve the readability of the manuscript;
- L390-395: this paragraph seems a bit odd for the data and materials section. The authors argued about the policy implications of their methodology which could be inside the discussion section of the paper;
- The raw data in Table 3 must be explained;
- Table 4 is a bit hard to read. P values are presented directly from the software. Instead of 2E-15, please use the classical representations such as 0.01, 0.05, or 0.10, etc;
- L473-476: Please, indicate those studies once again to remind for the readers;
- L487-488: Is the discovery of additional AEM peculiarities a surprising finding? If so, I would suggest rephrasing the sentence to make an emphasis on it;
- In general, the conclusion section is well-elaborated and meets the expectations; however, the findings of Tables 4 and 5 must be also incorporated in a more concrete and narrow style.
Minor comments:
- L5-6: Incomplete and improper ideas. Try to revise it please, or simply delete them;
- L6: “Ongoing crisis” – which one? I can guess about COVID19, but in the Russian case maybe it is also the crisis between the USA and Russia?
- L14-15: Which regression model? OLS, 2SLS, FMOLS, or DOLS? Please, specify not only in abstract but also in all relevant parts of the manuscript;
- L60-62: There is no need to write about the introduction section after readers reached the end of the introduction;
- L204-205: Unnecessary part. Please, either improve this paragraph to be more informative and engaging than its current form or remove it. Otherwise, it is just an orphan sentence.
Final opinion: In sum, I suggest publishing the paper after major revisions and corrections indicated in this peer review report. While the introduction section needs major improvements, the other sections of the paper are well-elaborated and well-written. I would say that, overall, the work achieves its main aims and goals by analyzing the Russian case in the context of interregional investment distribution employing the AEM model. The regression estimations supported the methodological considerations, as well as pointed to the statistical relationship among the variables of interest. The research hypothesis and research question must be incorporated with a more thorough caution. In other words, there is a need to improve them, while the modeling and mathematical intuition behind the research are clear and straightforward without any flaws and major shortcomings. Furthermore, I am sure that the paper might inspire new research in the case of the post-Soviet and developing countries related to interregional investment distribution employing the AEM model and regression estimation. Undoubtedly, the work fills the knowledge gap and addresses intriguing aspects of the interregional investment distribution in the case study of Russia. I would like to congratulate the authors.
Author Response
Response to Reviewer 3 Comments
Point 1:The abstract section does not include more narrowed and necessary insights about the findings of the paper. The authors argue about the significance and overall importance of the methodology and results; however, precise and accurate positioning of the results is necessary too;
Response 1:Is it challenging to convey full scope of our research with small word limit, we made some additions and alterations to the abstract, we tried to stay limited to minor changes as to not dilute abstract with overburdening numbers and facts, hopefully this changes will be to your liking.
Point 2:I strongly recommend to include the end-users of the findings of this paper at the end of the abstract section;
Response 2:End-users of the research were added to the abstract. Changes were limited to 3 main interest groups as to not inflate abstract word count. L25-27
Point 3: The citation density of the first two paragraphs of the introduction section is low. Without proper references, introductory sentences are just claims, not argumentation chains. I am sure authors can find proper papers and works to include about overall Russian case;
Response 3:Addinitional references were added to the introduction. Apart from research papers we added relevant reports made by The World Bank and Boston Consulting group. L34, L37, L41, L44.
Point 4: Starting from the third paragraph of the introduction section, the authors describe how the regional distribution of the investments is inequal and lags from each other. This situation must be linked to a more general phenomenon in the case of the Russian economy. For instance, Dutch disease and Natural Resource Curse theories widely capture why certain sectors and regions lag due to the booming sectors in a given economy. Therefore, the authors can develop the third paragraph even further by introducing a top-down approach. In other words, the link between the issues of interregional investment distribution and the overall macroeconomic situation might be connected to Dutch Disease. This was found in the other post-soviet countries. Please, refer to Niftiyev (2020) [https://www.econstor.eu/handle/10419/227485], Niftiyev 2021 [https://econpapers.repec.org/article/higecohse/2021_3a2_3a6.htm] and Sadik-Zada (2019) https://link.springer.com/article/10.1007/s13563-019-00202-6 where the Azerbaijani case was analyzed and Kazakhstan https://www.tandfonline.com/doi/abs/10.1080/14631377.2020.1745557 which is similar to the oil and gas-rich Russia;
Response 4:The problemindeed has an underlaying general phenomenon of Soviet-Russia transformation, which arenot widely covered or considered by western researchers. Historic practices of the Soviet union outlined future development of what Russia has today. Additional explanation and relevant research were added to the article. Additional information is concise as to not shift focus from primary research question. L56-L63
Point 5: L47-49: Some sort of brief literature references should be inserted here to describe and argue about the overall relevance of the applied methodology and research design;
Response 5:additional literature references were added. L55
Point 6: L59-59: Try to revise the hypothesis. The hypothesis must meet the widely accepted experience of hypothesis-writing. In its current form, it is unsatisfactory;
Response 6:hypothesis was revised and rewriten. Additionally we added research question that was formed to better suit main focus of the paper. L67-L75
Point 7: L70: I strongly suggest renaming Figure 1 to research design, rather than “methodology” and putting it into the data and materials section. Moreover, Figure 1 has not been explained. The text should reflect the essential ideas of the research design described by the flow chart;
Response 7:figure moved to “Materials and Methods” section, name is changed to research design. Additonal explanations were added explaining research deasign. L227-L233
Point 8: The literature review section is good, but some minor improvements are needed. Again, similar to the introduction section, some paragraphs of the literature review lack the proper quantity of citations. Moreover, I suggest replacing the paragraph between the lines of L143-148 from its current position, to the end of the section;
Response 8:Additional references were added to the “Literature section”. L151-155, L172,.
Paragrapgh from L143-148 was moved to the end of the section.
Point 9:Usually, at the end of the literature review sections, research questions are presented and research objectives are repeated. I strongly suggest doing so which could improve the readability of the manuscript;
Response 9:Research question and objectives appended at the end of literature section, research question was already established in the introduction section( along the hypothesis).
Point 10: L390-395: this paragraph seems a bit odd for the data and materials section. The authors argued about the policy implications of their methodology which could be inside the discussion section of the paper;
Response 10:paragraph was moved to discussion section. L537-L542
Point 11: The raw data in Table 3 must be explained;
Response 11:additional explanation was added to table 3, authors apended explanations as to why Moscow was added dispite it economicaly dwarfing any other region. L423-L430
Point 12: Table 4 is a bit hard to read. P values are presented directly from the software. Instead of 2E-15, please use the classical representations such as 0.01, 0.05, or 0.10, etc;
Response 12:values were changed to a more suitable format. L431-L432
Point 13:L473-476: Please, indicate those studies once again to remind for the readers;
Response 13:reference links were added L503
Point 14: L487-488: Is the discovery of additional AEM peculiarities a surprising finding? If so, I would suggest rephrasing the sentence to make an emphasis on it;
Response 14: AEM values are not that suprising and are well undertood, parts of the paragraph were reworked and removed to establish proper meaning.REMOVED
Point 15: In general, the conclusion section is well-elaborated and meets the expectations; however, the findings of Tables 4 and 5 must be also incorporated in a more concrete and narrow style.
Response 15:small changes were made to conclusionslinking results to findings in tables 4 and 5.L560-562
Point 16: L6: “Ongoing crisis” – which one? I can guess about COVID19, but in the Russian case maybe it is also the crisis between the USA and Russia?
Response 16:Clarifications were added as to what type of crisis authors ment( COVID). L9
Point 17: L14-15: Which regression model? OLS, 2SLS, FMOLS, or DOLS? Please, specify not only in abstract but also in all relevant parts of the manuscript;
Response 17:Authors meant paired linear regression, necessery changes were made throught the text. Changes were highlited.L16, L237, L278, L432, L433. L455, L464, L465,
Point 18: L60-62: There is no need to write about the introduction section after readers reached the end of the introduction;
Response 18:that part of the text was removed. REMOVED
Point 19: L204-205: Unnecessary part. Please, either improve this paragraph to be more informative and engaging than its current form or remove it. Otherwise, it is just an orphan sentence.
Response 19:Part was removed. REMOVED
Round 2
Reviewer 1 Report
The revised paper included the developed methods and results.
Author Response
We would like to thank you for your thorough review, it was our pleasure to work with someone so dedicated.
Reviewer 3 Report
Unfortunately, you have not followed fully my comments. Especially with regards to referring to relevant literature. Please scrutinize my latest review report again and augment the literature by the authors I mentioned. Afterwards the paper could be published.
Author Response
The authors would like to thank the reviewer for his/her comments that helped to improve the manuscript. We consulted your previous comments and integrated literature and aforementioned theories to introduction. L59-L63, L615-L627
Round 3
Reviewer 3 Report
The paper can be accepted now.